
# Quantum chaos in a harmonic waveguide with scatterers

Vladimir A. Yurovsky⋆

School of Chemistry, Tel Aviv University, 6997801 Tel Aviv, Israel

⋆ volodia@post.tau.ac.il

## Abstract

A set of zero-range scatterers along its axis lifts the integrability of a harmonic waveguide. Effective solution of the Schrödinger equation for this model is possible due to the separable nature of the scatterers and millions of eigenstates can be calculated using modest computational resources. Integrability-chaos transition can be explored as the model chaoticity increases with the number of scatterers and their strengths. The regime of complete quantum chaos and eigenstate thermalization can be approached with 32 scatterers. This is confirmed by properties of energy spectra, the inverse participation ratio, and fluctuations of observable expectation values.

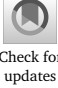

# 1  Introduction

Completely-chaotic systems have impredictable ergodic trajectories (see [1]) and their average properties can be described by the Gibbs statistical ensemble [2]. In quantum systems, the statistical description is a consequence of the eigenstate thermalization hypothesis (ETH), introduced in [3,4] (see also [5,6], the experimental work [7], the review [8] and the references therein). Energy spectra of completely-chaotic systems follow Wigner-Dyson statistics with a dip at small level spacings [9–11]. In contrast, trajectories of classical integrable systems are completely predictable and, according to the Kolmogorov-Arnold-Moser theorem, this property remains even when a weak integrability-breaking perturbation is applied [1]. Quantum systems demonstrate similar properties (see, e.g., [12–14]). A statistical description by the generalized Gibbs ensemble [15–20] is applicable to the final state of integrable system relaxation. The Poisson statistics of integrable system energy spectra has no dip at small spacings [9–11].

However, a generic system is not completely chaotic nor integrable (see examples in [21–52]). Certain incompletely-chaotic systems — the systems with no selection rules — relax to a state whose properties are governed by the inverse participation ratio (IPR) [28,31]. Inverse of this parameter estimates the number of integrable system eigenstates comprising the non-integrable one. IPR ranges from 0 for completely-chaotic systems to 1 for integrable ones. Then it can serve as a measure of the system's chaoticity [53]. IPR also governs fluctuations of eigenstate expectation values [32]. The energy-spectrum statistics of incompletely-chaotic systems lie between the Wigner-Dyson and Poisson ones. Certain systems demonstrate the Šeba statistics [22].

The most obvious objects of chaotic property simulation are lattice systems. However, they have a finite Hilbert space and its dimension is restricted due to computational difficulties (complexity of lattice system simulations increases as a high power of the lattice site number and exponentially with the number of particles). Then, on increase of the system chaoticity, each eigenstate can fill the full Hilbert space. A system with infinite Hilbert space — the Sinai type billiard — was analyzed in [54], where $\sim 3 \times 10^5$ eigenstates were calculated. However, chaoticity of such billiard cannot be tuned.

The present model — a particle in a harmonic waveguide with zero-range scatterers along its axis — has an infinite Hilbert space. As the scatterers are a particular case of independent perturbations [55], IPR should be inversely proportional to the number of scatterers. The model chaoticity can also be tuned by the scatterer strengths. This model was already used in [55] for numerical confirmation of the general relations between properties of wavefunctions and the number of scatterers. The present paper is devoted exclusively to the harmonic waveguide with scatterers and analyzes properties of wavefunctions for weak perturbations and for additional models, as well as properties of energy spectra.

Since a zero-range scatterer is a particular case of separable interactions, the present model belongs to systems with high-rank separable perturbations [56]. Energy spectra of several physical systems of such type have already been considered. They are the flat rectangular billiards — generalization of the Šeba billiard [21] — with 1-3 [23], 6 [24], and 2 [46] scatterers. Theoretical predictions for a single scatterer in a harmonic potential were compared to experiments [57]. Series of separable interactions can also approximate the dipole-dipole ones [58,59]. Energy spectra of two dipolar particles in a harmonic trap were calculated [60] using such expansion. An advantage of systems with separable rank-$s$ interactions is that calculations require diagonalization of a $s \times s$ matrix, (cf. to $\alpha \times \alpha$ matrix in the direct diagonalization method for $\alpha$ eigenstates). In addition, the present model allows an analytical summation over axial states. Then the system properties are calculated here for millions of eigenstates.

The paper has the following organization. The model is described in Sec. (2). Section (3) analyzes the energy spectra statistics. Properties of wavefunctions, including expectation value fluctuations and IPR, are presented in Sec. 4. Appendices provide derivation details and additional technical information.

A system of units in which Planck's constant is $\hbar = 1$ is used below.

## 2 The model

The Hamiltonian of a particle with the mass $m$ in an axially-symmetric harmonic waveguide with the transverse frequency $\omega_\perp$ contains the kinetic and potential energies,

$$\hat{H}_0 = \frac{1}{2m}\left[\left(\frac{1}{i}\frac{\partial}{\partial z} - A\right)^2 - \triangle_\rho\right] + \frac{m\omega_\perp^2 \rho^2}{2}. \tag{1}$$

Here $z$ is the axial coordinate, $\rho = \sqrt{x^2 + y^2}$ is the transverse radius, $\triangle_\rho$ is the transverse Laplacian, and $A$ is the vector potential (its role will be discussed below).

Integrability of the perturbed Hamiltonian

$$\hat{H}_s = \hat{H}_0 + \sum_{s'=1}^{s} \hat{V}_{s'}, \tag{2}$$

is lifted by the zero-range scatterers

$$\hat{V}_{s'} = V_{s'}\delta_{reg}(\mathbf{r} - \mathbf{R}_{s'}), \tag{3}$$

where $\delta_{reg}$ is the Fermi-Huang pseudopotential and the scatterers are located along the waveguide axis, i.e., their positions $\mathbf{R}_{s'} = (0,0,z_{s'})$ have zero transverse coordinates. The scatterers are numbered from left to right ($z_{s'} > z_{s''}$ if $s' > s''$). The model is restricted in the sector of the axially-symmetric states, as other states vanish at the waveguide axis, and, therefore, are not affected by the scatterers. Then the eigenstates of $\hat{H}_0$, labeled by the axial $l$ and radial $n \geq 0$ quantum numbers, are $\langle\rho, z|nl\rangle = \langle\rho|n\rangle\langle z|l\rangle$ with the radial wavefunctions

$$\langle\rho|n\rangle = \frac{1}{\sqrt{\pi}a_\perp}L_n^{(0)}((\rho/a_\perp)^2)\exp(-(\rho/a_\perp)^2/2). \tag{4}$$

Here $a_\perp = (m\omega_\perp)^{-1/2}$ is the transverse oscillator range and $L_n^{(0)}$ are the Laguerrre polynomials (see [61]). The discrete energy spectrum is provided either by the periodic boundary conditions (PBC) $\langle z + L|l\rangle = \langle z|l\rangle$, or by the hard-wall box (HWB) $\langle z = L|l\rangle = \langle z = 0|l\rangle = 0$. Then the axial wavefunctions are either

$$\langle z|l\rangle = L^{-1/2}e^{2i\pi l\zeta}, \tag{5}$$

with $-\infty < l < \infty$ and $\zeta = z/L$ for PBC or

$$\langle z|l\rangle = (2/L)^{1/2}\sin\pi l\zeta, \tag{6}$$

with $1 \leq l < \infty$ for HWB.

The particular case of PBC with a single scatterer was considered in [26–29].

For PBC, the eigenstate $|nl\rangle$ of $\hat{H}_0$ has the eigenenergy

$$E_{nl} = \frac{2}{mL^2}\varepsilon_{nl} + \omega_\perp, \quad \varepsilon_{nl} = \lambda n + \pi^2(l - l_0)^2, \tag{7}$$

where $\lambda = (L/a_\perp)^2$ characterizes the aspect ratio and $l_0 = LA/(2\pi)$ is the scaled vector poten-
tial. If $A = 0$, the inversion (P) invariance of the Hamiltonian $\hat{H}_0$ leads to the degeneracy of
the energies $E_{nl}$ and $E_{n-l}$. This degeneracy can be lifted by any P-noninvariant perturbation.
The vector potential lifts it as well, with no effect on the simple wavefunctions (5), though the
Hamiltonian losses the time-reversal (T) invariance.

Four kinds of the model are considered here. The first three kinds correspond to PBC. The
first, non-symmetric, model has $A \neq 0$ and is T-noninvariant. The scatterer positions

$$z_1 = 0, \quad z_{s'} = (s' - 1 + \delta_{s'})L/s \quad (s' > 1), \tag{8}$$

form irregular sequence due to random shifts $-0.25 \le \delta_{s'} < 0.25$. The shifts are calculated
once for each number of scatterers and there is no average over the shifts. In the second,
symmetric, model with $z_{s-s'+1} = z_s - z_{s'} + z_1$ for $s' > s/2$, the scatterer positions are invariant
over inversion under $(z_1 + z_s)/2$. This inversion changes the sign of the term $(i/m)A\partial/\partial z$ in the
Hamiltonian $\hat{H}_0$. This sign is also changed by the time-reversal (complex conjugation). Then
the symmetric model with equal $V_{s'}$ is PT-invariant. The third, T-invariant, model has $A = 0$
and the same scatterer positions as the non-symmetric one. Only this model has a degenerate
energy spectrum of the integrable Hamiltonian. The fourth, box, model corresponds to HWB.
The scatterer positions are $z_{s'} = (s' + \delta_{s'})L/(s + 1)$. Although $A = 0$, the energy spectrum

$$\varepsilon_{nl} = \lambda n + \frac{\pi^2}{4}l^2, \tag{9}$$

is non-degenerate as $l$ is positive.

Together, the four kinds of the model cover different symmetries of the Hamiltonian (T-
invariant, PT-invariant, and non-symmetric), as well as different boundary conditions (PBC
and HWB).

The eigenstates of the non-integrable system $|\alpha\rangle$, solutions to the Schrödinger equation
$\hat{H}|\alpha\rangle = E_\alpha|\alpha\rangle$, are labeled in the increasing order of the eigenenergies $E_\alpha$. Expansion over the
integrable system eigenstates $|nl\rangle$ transforms the Schrödinger equation to the form

$$|\alpha\rangle = \sum_{n,l} \frac{|nl\rangle \langle nl|}{E_\alpha - E_{nl}} \sum_{s'=1}^{s} \hat{V}_{s'}|\alpha\rangle. \tag{10}$$

According to (3)

$$\langle nl|\hat{V}_{s'}|\alpha\rangle = V_{s'}\langle nl|\mathbf{R}_{s'}\rangle \langle\mathbf{R}_{s'}|\alpha\rangle_{reg}, \tag{11}$$

where the value of the regular part of $|\alpha\rangle$ at $\mathbf{R}_{s'}$ is

$$\langle\mathbf{R}_{s'}|\alpha\rangle_{reg} = \frac{\partial}{\partial r}[r\langle\mathbf{r}|\alpha\rangle]_{\mathbf{r}=\mathbf{R}_{s'}} = \frac{\partial}{\partial z}[z\langle 0,0,z|\alpha\rangle]_{z=z_{s'}}. \tag{12}$$

The last equality above follows from the spherical symmetry of $\langle\mathbf{r}|\alpha\rangle$ in the vicinity of $\mathbf{R}_{s'}$ [62].
As a result, we get the following system of linear equations for $\langle\mathbf{R}_{s'}|\alpha\rangle_{reg}$

$$\langle\mathbf{R}_{s'}|\alpha\rangle_{reg} = \sum_{s''=1}^{s} V_{s''}\frac{\partial}{\partial z}\left[z\sum_{n,l}\frac{\langle 0,0,z|nl\rangle \langle nl|0,0,z_{s''}\rangle}{E_\alpha - E_{nl}}\right]_{z=z_{s'}}\langle\mathbf{R}_{s''}|\alpha\rangle_{reg}. \tag{13}$$

For the wavefunctions (4) and (5) or (6) and energies (7) or (9) the sum over $l$ above can be
calculated analytically (see Appendix A). Then the system (13) attains the form

$$\sum_{s''=1}^{s} S_{s's''}(\varepsilon)\langle\mathbf{R}_{s''}|\alpha\rangle_{reg} = 0, \tag{14}$$

with

$$S_{s's''}(\varepsilon) = \frac{V_{s''}}{V_0}\sqrt{\lambda}\sum_{n=0}^{\infty} T_n(\zeta_{s'},\zeta_{s''}) \quad (s' > s''), \qquad S_{s''s'}(\varepsilon) = S_{s's''}^*(\varepsilon), \tag{15}$$

$$S_{s's'}(\varepsilon) = \frac{V_{s'}}{V_0}\left[\sqrt{\lambda}\left(\sum_{n=0}^{[\varepsilon/\lambda]} T_n(\zeta_{s'},\zeta_{s'}) + \sum_{n=[\varepsilon/\lambda]+1}^{\infty} T_n^{reg}(\zeta_{s'})\right) - \zeta\left(\frac{1}{2},\left[\frac{\varepsilon}{\lambda}\right]+1-\frac{\varepsilon}{\lambda}\right)\right] - 1. \tag{16}$$

Here [] denote the integer part, $\zeta_{s'} = z_{s'}/L$, $\zeta(.,.)$ is the Hurwitz zeta function (see [61]), $V_0 = 2\pi a_\perp/m$ is the scale of the interaction strength, and the summands $T_n(\zeta_{s'},\zeta_{s''})$ and $T_n^{reg}$ are given in Appendix A for each kind of the model. Due to arrangement of scatterers, only $T_n(\zeta_{s'},\zeta_{s''})$ with $\zeta_{s'} \geq \zeta_{s''}$ have to be calculated. $T_n(\zeta_{s'},\zeta_{s'})$ and $T_n^{reg}$ are always real functions. If $A = 0$, $T_n(\zeta_{s'},\zeta_{s''})$, as well as the matrix $S_{s's''}(\varepsilon)$, is real, and $S_{s's''}(\varepsilon)$ is symmetric.

The system (14) has a non-trivial solution at $\varepsilon = \varepsilon_\alpha \equiv mL^2(E_\alpha - \omega_\perp)/2$ where an eigenvalue of its matrix has a root as a function of $\varepsilon$. The matrix $S_{s's''}(\varepsilon)$ has poles at $\varepsilon = \varepsilon_{nl}$, as it is seen from (13). Then the eigenvalues can have poles at $\varepsilon = \varepsilon_{nl}$ as well. Between these poles each eigenvalue is a monotonic function of $\varepsilon$, as demonstrated by direct calculations (see Appendix B). Than all eigenenergies $\varepsilon_\alpha$ in each interval between neighboring $\varepsilon_{nl}$ can be calculated as roots of $s$ eigenvalues. Although the eigenvalue monotonicity was not proved exactly, this algorithm provides the number of eigenenergies $\varepsilon_\alpha$ which differs from the number of $\varepsilon_{nl}$ in the same energy interval by not more than $s$. It is an evidence that no eigenenergies $\varepsilon_\alpha$ are lost.

The terms in the sums over $n$ in Eqs. (15) and (16) decay exponentially when $n > \varepsilon_\alpha/\lambda$ (see Appendix A). Thus, the calculation of the system (14) matrix requires $\propto s^2\alpha^{2/3}$ operations since $\varepsilon_\alpha \propto \alpha^{2/3}$ [see Eq. (19) below], while its solution requires $\propto s^3$ operations. Then, if $s \ll \alpha^{2/3}$, calculation of $\alpha$ eigenenergies requires $\propto s^2\alpha^{5/3}$ operations — much less than $\alpha^3$ operations in the direct diagonalization method.

There seems to be no fundamental obstacle for experimental realization of the present model. In the case of cold trapped atoms, atoms of other kind in optical tweezers might play the role of scatterers, and the interaction strength might be tuned by a Feshbach resonance. T-noninvariant models might be realized with trapped ions in a magnetic field. In optics, optical defects might work as scatterers [63] for photons in an optical cavity or waveguide (see also [64, 65] and the references therein). The PBC models might be realized with circular atomic or optical waveguides.

## 3 Statistics of energy spectra

The differences in energy spectra between integrable and chaotic systems were the first distinctive properties of quantum chaos (see [9–11]). These properties are defined in terms of the unfolded energy $\bar{\alpha}(\varepsilon_\alpha)$ — the smooth part of the dependence $\alpha(\varepsilon_\alpha)$. For the present model, the unfolding function is the same as for the underlying integrable system. The number of states below the scaled energy $\varepsilon$ is the staircase function

$$\alpha(\varepsilon) = \sum_{n,l} \theta(\varepsilon - \varepsilon_{nl}). \tag{17}$$

For PBC, using Eq. (7) for $\varepsilon_{nl}$, we have

$$\alpha(\varepsilon) = \sum_{l=-\infty}^{\infty} \left\{[(\varepsilon - \pi^2(l-l_0)^2)/\lambda]+1\right\}\theta((\varepsilon - \pi^2(l-l_0)^2)/\lambda+1). \tag{18}$$

The smooth part is extracted by replacing the integer part $[x]$ with $x - 1/2$. The limits of the sum over $l$, $[l_0 \pm \sqrt{\varepsilon}/\pi]$, are replaced in the same way. As a result, we get

$$\bar{\alpha}(\varepsilon) = \frac{4}{3\pi\lambda}\varepsilon^{3/2} + \left(\frac{1}{\pi} + \frac{\pi}{6\lambda}\right)\varepsilon^{1/2}. \tag{19}$$

The $\varepsilon$-independent terms are dropped here, since only differences between $\bar{\alpha}(\varepsilon_\alpha)$ appear in the following expressions. Similar expression is obtained for the HWB model

$$\bar{\alpha}(\varepsilon) = \frac{4}{3\pi\lambda}\varepsilon^{3/2} - \frac{\varepsilon}{2\lambda} + \left(\frac{1}{\pi} + \frac{\pi}{24\lambda}\right)\varepsilon^{1/2}. \tag{20}$$

The first property of the energy spectrum considered here is the nearest-neighbor distribution (NND) — the density of probability to have the given value of the unfolded energy difference $\bar{\alpha} = \bar{\alpha}(\varepsilon_\alpha) - \bar{\alpha}(\varepsilon_{\alpha-1})$ between the neighboring energy levels [9–11]. Integrable systems have the Poisson NND,

$$w_{Pois}(\bar{\alpha}) = e^{-\bar{\alpha}}, \tag{21}$$

while completely-chaotic ones have the Wigner-Dyson distributions for Gaussian ensembles of random orthogonal matrices (GOE)

$$w_{GOE}(\bar{\alpha}) = \frac{\pi}{2}\bar{\alpha}\exp\left(-\frac{\pi}{4}\bar{\alpha}^2\right), \tag{22}$$

and unitary matrices (GUE)

$$w_{GUE}(\bar{\alpha}) = \frac{32}{\pi^2}\bar{\alpha}^2\exp\left(-\frac{4}{\pi}\bar{\alpha}^2\right), \tag{23}$$

in the cases of T-invariant and T-noninvariant systems, respectively.

The Šeba NND

$$w_{Seba}(\bar{\alpha}) = A_{Seba}\bar{\alpha}\exp\left(-B_{Seba}\bar{\alpha} - \frac{A_{Seba}}{B_{Seba}^2}\left[1 - e^{-B_{Seba}\bar{\alpha}}(B_{Seba}\bar{\alpha} + 1)\right]\right), \tag{24}$$

with $A_{Seba} \approx 2.1266$ and $B_{Seba} \approx 0.3481$ was obtained [22] for certain incompletely-chaotic systems.

All states of a non-integrable system correspond to the same symmetry and then their energies demonstrate repulsion. Then NND (22),(23), and (24) of non-integrable systems vanish at the zero level spacing and decrease approaching this point. The integrable system states of different symmetry can be energy degenerate, and then NND (21) decreases exponentially with the level spacing. For non-integrable systems NND decreases at large spacing too, although completely-chaotic systems are characterized by Gaussian decrease [see Eqs. (22) and (23)], while the Šeba NND (24) decreases exponentially.

Another property of energy spectra is the spectral rigidity $\Delta_3(\Delta\bar{\alpha})$ — the least-square deviation of the staircase function $\alpha(\varepsilon)$ from the best fit to a straight line on a given interval of the unfolded energy $\Delta\bar{\alpha}$. The spectral rigidity for integrable and completely-chaotic (T-invariant and T-noninvariant) systems are given, respectively, by [9–11]

$$\begin{aligned}\Delta_3^{Pois}(\Delta\bar{\alpha}) &= \Delta\bar{\alpha}/15, \\ \Delta_3^{GOE}(\Delta\bar{\alpha}) &= \frac{1}{\pi^2}\left(\ln\Delta\bar{\alpha} + \ln 2\pi + \gamma_{Eul} - \frac{5}{4} - \frac{\pi^2}{8}\right), \\ \Delta_3^{GUE}(\Delta\bar{\alpha}) &= \frac{1}{2\pi^2}\left(\ln\Delta\bar{\alpha} + \ln 2\pi + \gamma_{Eul} - \frac{5}{4}\right),\end{aligned} \tag{25}$$

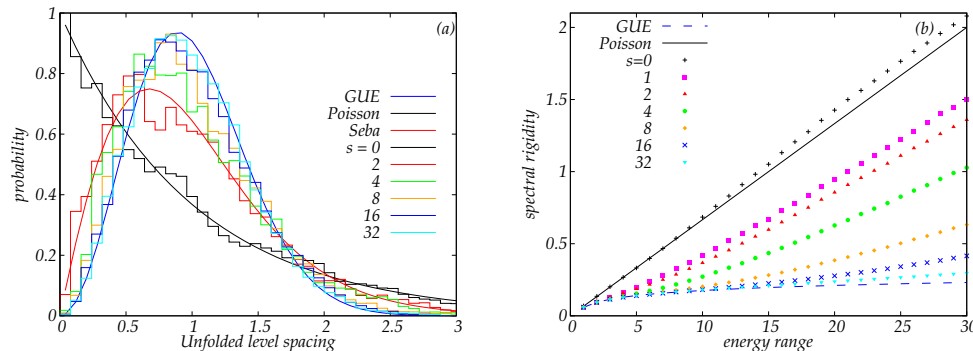

Figure 1: Near-neighbor distribution (a) and the spectral rigidity (b) for the non-symmetric model with different numbers of scatterers in the unitary regime .

where $\gamma_{Eul} \approx 0.5772$ is the Euler's constant [61].

The energy spectrum properties are calculated below for the four kinds of the models. The parameters $l_0 = 0.25 - e^{-4} \approx 0.232$ (for the T-noninvariant models) and $\lambda = \pi^3(1+\sqrt{5}) \approx 100$ are expressed in terms of transcendent numbers [$(1+\sqrt{5})/2$ is the golden ratio]. Most of the results are obtained for $10^6$ eigenstates in the unitary regime, $V_{s'} = 10^6 V_0$ for all scatterers.

Figure 1(a) shows NND calculated for the non-symmetric model with different numbers of scatterers in the unitary regime. For $s = 2$, NND follows the Šeba plot, as well as for the case of $s = 1$ considered in [27]. When the number of scatterers increases, NND tends to the GUE prediction and approaches it at $s = 32$. GUE is approached as the model is T-noninvariant. The calculated spectral rigidity [see Fig. 1(b)] demonstrates the same tendency.

For the symmetric model (see Fig. 2), NND for $s = 2$ again follows the Šeba predictions (indeed, the case of two scatterers of the same strength is always P-invariant). However, at $s = 32$, NND and spectral rigidity approach the GOE predictions, although the system is T-noninvariant. It is a consequence of the real matrix of the interaction with scatterers

$$\left\langle n'l' \left| \sum_{s'=1}^{s} \hat{V}_{s'} \right| nl \right\rangle = \frac{V_1}{\pi L a_\perp^2} \sum_{s'=1}^{s} \cos 2\pi(l-l')\tilde{\zeta}_{s'}, \tag{26}$$

obtained when the $z$ coordinate origin is shifted to $(z_1 + z_s)/2$, such that $\tilde{\zeta}_{s'} = \zeta_{s'} - (\zeta_1 + \zeta_s)/2$. The real matrix should be described by GOE, like in T-invariant systems.

NND and spectral rigidity for the T-invariant model are shown in Fig. 3. Now Šeba and GOE NND are approached only at $s = 32$ and $s = 64$, respectively. For $s = 3$ both NND and

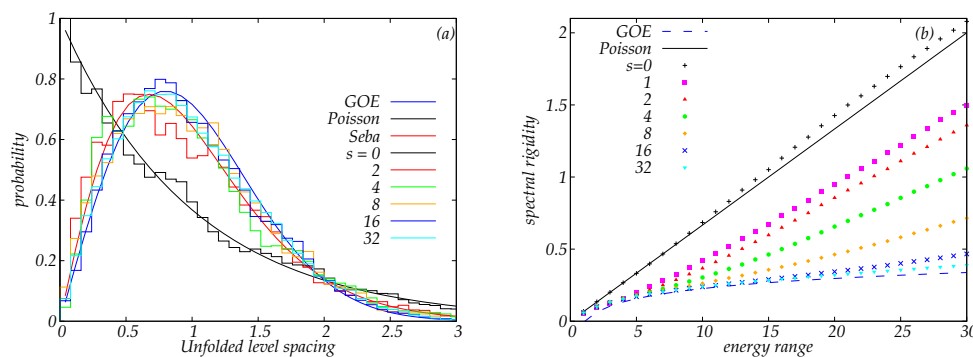

Figure 2: Near-neighbor distribution (a) and the spectral rigidity (b) for the symmetric model with different numbers of scatterers in the unitary regime .

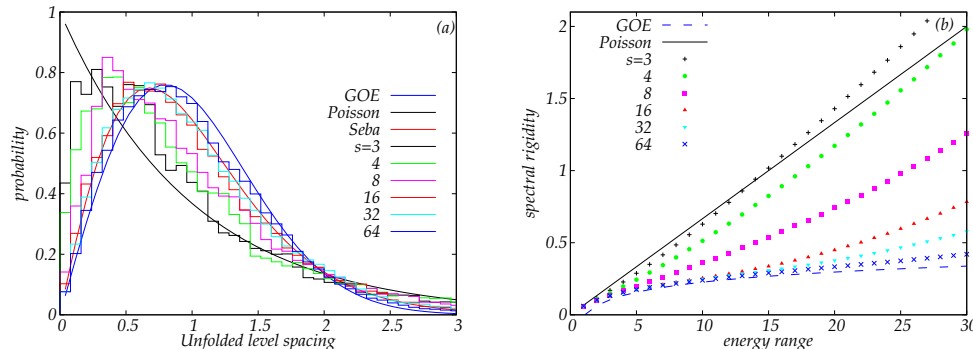

Figure 3: Near-neighbor distribution (a) and the spectral rigidity (b) for the T-invariant PBC model with different numbers of scatterers in the unitary regime .

spectral rigidity are close to the Poisson predictions. Then, this model is less chaotic than the T-noninvariant ones where the Šeba and Wigner-Dyson statistics are approached at $s = 2$ and $s = 32$, respectively. This may be related to degeneracy of the integrable system energy spectrum for the T-invariant model.

This assumption is confirmed by the NND and spectral rigidity for the HWB model (see Fig. 4). This model with non-degenerate energy spectrum is more chaotic than the PBC T-invariant one, as now Šeba and GOE predictions are approaching at $s = 16$ and $s = 32$, respectively. Then, this model is less chaotic than the T-noninvariant PBC ones. There is also a noticeable difference between these models in the statistics of integrable system energy spectra — for the HWB model NND at small spacings and spectral rigidity are below the Poisson predictions.

Thus, for all kinds of the model the statistics tend to the Wigner-Dyson predictions on increase of the number of scatterers. This agrees with the behavior of spectral rigidity of flat 2D billiards [23]. However, the present model does not demonstrate another property of the 2D flat billiards — the shifting toward Poisson statistics at higher energy [23]. It is clearly shown in Fig. 5, where the plots for different energy regions are close together and do not demonstrate a systematic dependence on the energy. This difference is related to the nature of the logarithmic asymptotic freedom revealed in [23]. This effect is caused by the decreased effective interaction strength $v_{\text{eff}} \sim 1/\ln \varepsilon$ (see Eq. (19) in [23]), while the characteristic energy level separation $\partial \varepsilon_\alpha / \partial \alpha$ is independent of the energy for 2D billiards with $\varepsilon_\alpha \propto \alpha$. In contrast, if $\varepsilon_\alpha \propto \alpha^\gamma$ ($\gamma \neq 1$), the derivation [23] would lead to $v_{\text{eff}} \propto \varepsilon^{1-1/\gamma}$, while $\partial \varepsilon_\alpha / \partial \alpha \propto \varepsilon^{1-1/\gamma}$ has the same energy dependence and ratio of the effective interaction strength to energy level separation is independent of energy. Therefore, the logarithmic asymptotic freedom does not

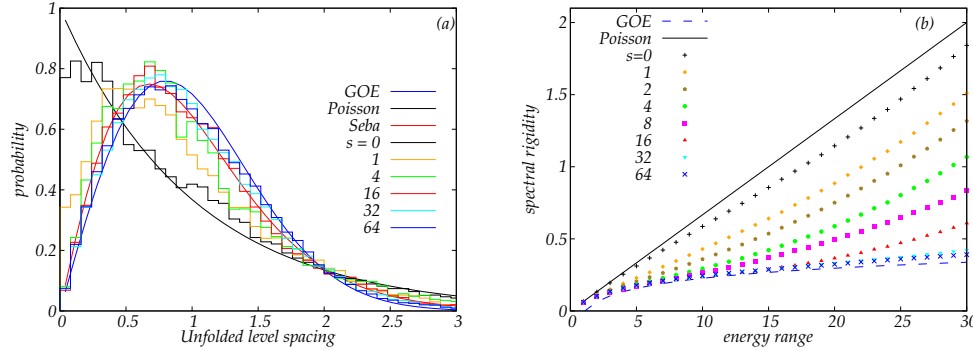

Figure 4: Near-neighbor distribution (a) and the spectral rigidity (b) for the HWB model with different numbers of scatterers in the unitary regime .

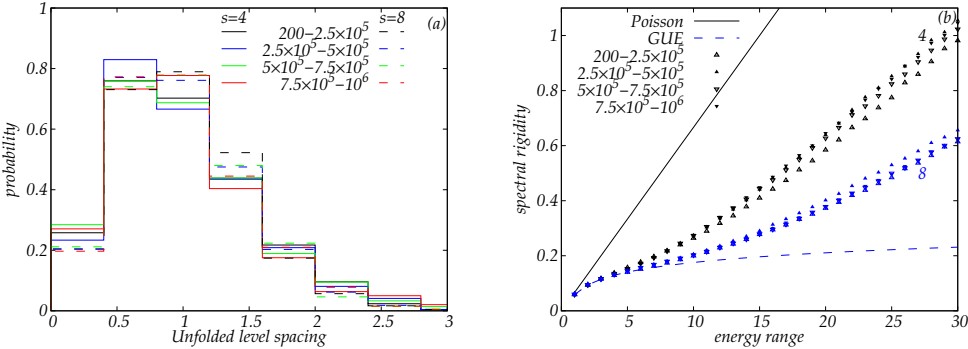

Figure 5: Near-neighbor distribution (a) and the spectral rigidity (b) for the non-symmetric model with 4 and 8 scatterers in the unitary regime at various regions of the non-integrable system eigenstate labels $\alpha$.

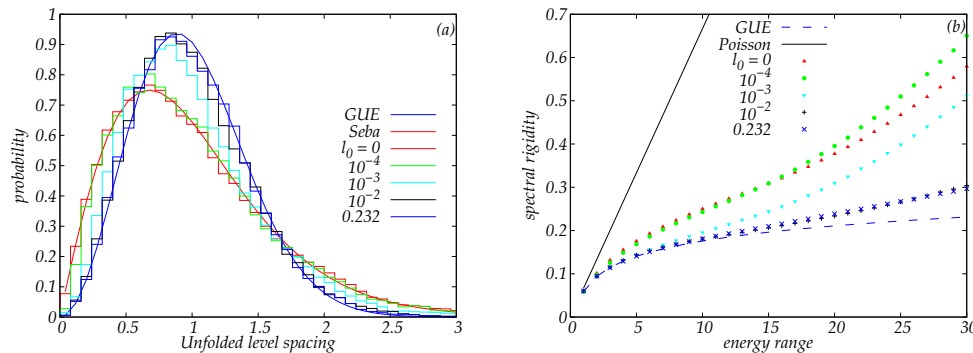

Figure 6: Near-neighbor distribution (a) and the spectral rigidity (b) for the non-symmetric model with 32 scatterers for various values of the scaled vector potential $l_0$.

appear in the present model with $\varepsilon_\alpha \propto \alpha^{2/3}$ as well as in generic systems with $\varepsilon_\alpha \propto \alpha^\gamma$ ($\gamma \neq 1$), being a specific property of 2D billiards.

The transition between the T-invariant and non-symmetric models due to the change of the vector potential is demonstrated in Fig. 6. The GUE and Šeba statistics take place at $l_0 < 10^{-4}$ and $l_0 > 10^{-2}$, respectively.

The system chaoticity depends also on the scatterer strength $V_{s'}$. NND approaches this

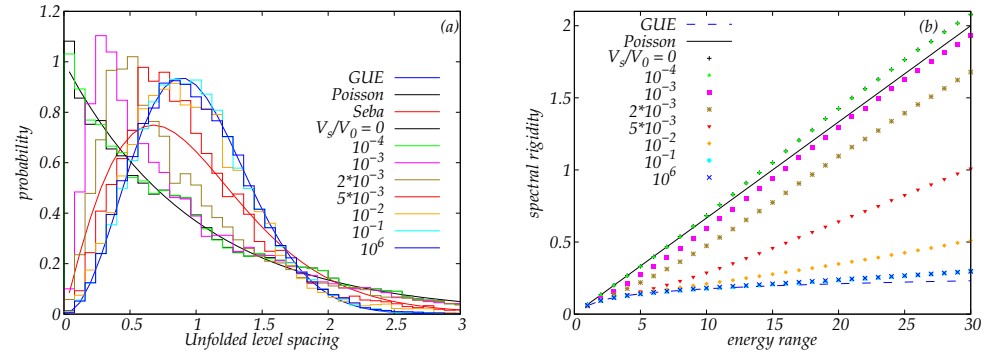

Figure 7: Near-neighbor distribution (a) and the spectral rigidity (b) for the non-symmetric model with 32 scatterers for various scatterer strengths.

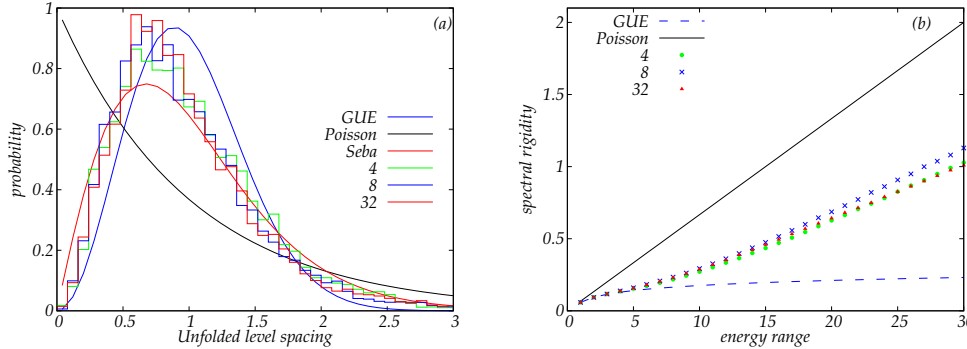

Figure 8: Near-neighbor distribution (a) and the spectral rigidity (b) for the non-symmetric model with 4, 8, and 32 scatterers at $V_{s'}/V_0 = 10^{-1}$, $10^{-2}$, and $5 \times 10^{-3}$, respectively.

unitary regime already at $V_{s'} = 10^{-1}V_0$, as Fig. 7(a) shows. For $V_{s'} = 10^{-4}V_0$ NND almost coincides with the integrable system one. Spectral rigidity demonstrates the same behavior (see Fig. 7(b)).

Thus, the system's chaotic properties depend on two parameters: the number of scatterers and their strengths. Interaction of these parameters is illustrated by Fig. 8, which demonstrates that the NND and spectral rigidity dependencies in the unitary regime for 4 scatterers are approached at $V_{s'} = 10^{-2}V_0$ and $V_{s'} = 5 \times 10^{-3}V_0$ for 8 and 32 scatterers, respectively.

Figure 9 shows dependence of the system statistics on the scatterer locations. All non-symmetric cases (1 and 2, corresponding to different sets of the random shifts $\delta_{s'}$ in (8), and 3, where $\zeta_1 = 0$ and $\zeta_{s'}$ with $s' > 1$ are chosen randomly from the interval $[0, 1]$ and sorted) provide close results approaching the GUE predictions. The plots for the symmetric distribution are clearly different and approach GOE predictions (see the discussion above).

If the scatterer positions form a periodic sequence, $\zeta_{s'} = (s' - 1)/s$ and $V_{s'}$ is constant, the picture is completely different. In this case, according to the Bloch's theorem, the eigenstate can be expressed as $\langle \rho, z | \alpha \rangle = \langle \rho, z | \alpha_p \rangle \exp(ipz)$. The $L$-periodicity plays the role of the Born-von Karman boundary conditions, leading to the discrete spectrum of the quasimomentum $p = 2\pi k_p/L$ with integer $k_p$. The function $\langle \rho, z | \alpha_p \rangle$ has the period $L/s$ and satisfies the Schrödinger equation with single scatterer

$$\left( \hat{H}_0(A - p) + \hat{V}_1 \right) |\alpha_p\rangle = E_{\alpha_p} |\alpha_p\rangle . \tag{27}$$

Here the integrable Hamiltonian $\hat{H}_0(A - p)$ of the form (1) contains the vector potential $A - p$.

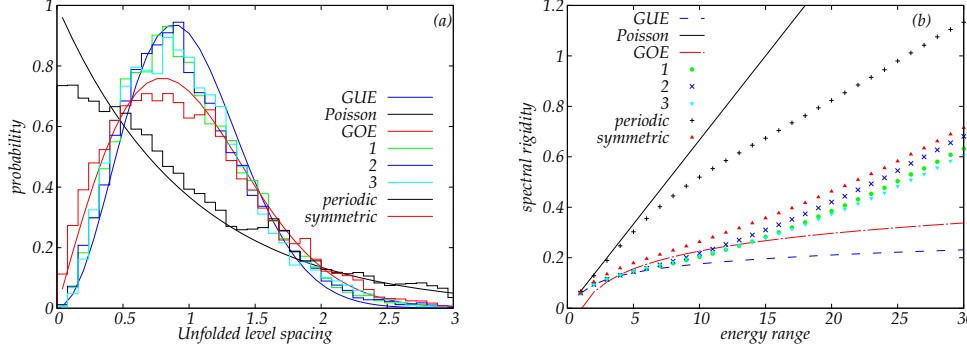

Figure 9: Near-neighbor distribution (a) and the spectral rigidity (b) for the non-symmetric model with 8 scatterers for various scatterer locations.

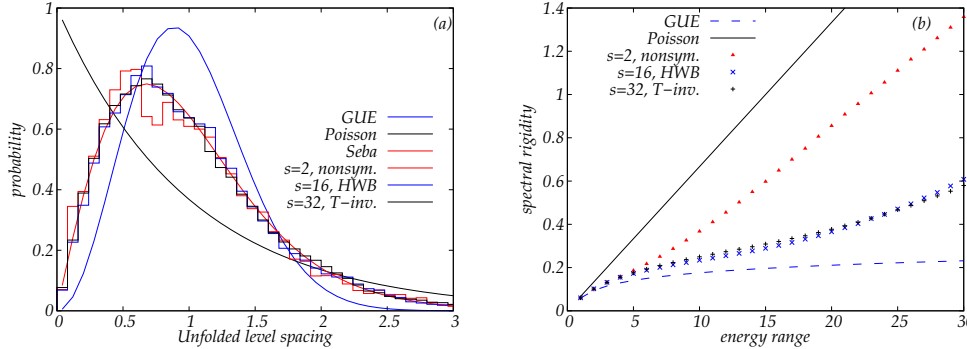

Figure 10: Near-neighbor distributions (a) which are close to the Šeba one for various models and the corresponding spectral rigidity (b).

Therefore, the total energy spectrum is a superposition of $s$ spectra of the one-scatterer systems with scaled vector potentials $l_0 - k_p$ ($k_p + s$ gives the same result as $k_p$). This is the reason (see [9]) why NND for the periodic case does not have a dip at small spacings and both NND and spectral rigidity are close to the Poisson predictions.

The random matrix theory [9–11] predicts universal spectral rigidity plots corresponding to the Poisson, GOE, and GUE NNDs. However, the Šeba NND can correspond to various spectral rigidity plots, as it is shown in Fig. 10. It is worth noting that the plots for the T-invariant PBC and HWB models are close together, while the one for the T-noninvariant model is completely different.

Statistics of energy spectra can be also characterized by average level spacing ratio [66, 67] which increases with the system chaoticity. In the present case (see Appendix C), this monotonic increase takes place only in the vicinity of the Poisson statistics. However, the average level spacing ratio becomes almost the same for the Šeba and GOE statistics and has strong fluctuations on the transition between them. This may be related to the small number of the degrees of freedom in the present models compared to many-body models, where the level spacing ratio is generally used. An additional advantage of the level spacing ratio is that unfolding the spectrum is not required. However, this advantage is not essential for the present models as the unfolding functions are well defined. For these reasons, the level spacing ratio is not used here.

## 4 Properties of wavefunctions

Possibility of statistical description of quantum-chaotic systems is based, through ETH, on properties of their wavefunctions. The number of integrable system eigenstates comprising the non-integrable one is characterized by the number of principal components (NPC) $\eta^{-1}$, where $\eta = \sum_{nl} |\langle n,l|\alpha\rangle|^4$ is IPR. Equations (10) and (11) allow us to express the expansion coefficients here in the form

$$\langle n,l|\alpha\rangle = \sqrt{\mathcal{N}_\alpha} \frac{1}{\varepsilon_\alpha - \varepsilon_{nl}} \sum_{s'=1}^{s} \frac{V_{s'}}{V_0} e^{-2i\pi l \zeta_{s'}} \langle \mathbf{R}_{s'}|\alpha\rangle_{reg}, \qquad (28)$$

where $\langle \mathbf{R}_{s'}|\alpha\rangle_{reg}$ are solutions to the system (14) and the normalization factor $\mathcal{N}_\alpha$ is determined by the normalization condition $\sum_{nl} |\langle n,l|\alpha\rangle|^2 = 1$. For the energies (7) the sums over

$n$ here and in IPR can be expressed in terms of the Hurwitz zeta functions (see [61])

$$\sum_{n=0}^{\infty} \frac{1}{(\varepsilon_\alpha - \varepsilon_{nl})^k} = \frac{(-1)^k}{\lambda^k} \zeta(k, q_l), \tag{29}$$

where

$$q_l = \frac{\pi^2 (l - l_0)^2 - \varepsilon_\alpha}{\lambda}. \tag{30}$$

Then the normalization condition takes the form $\sum_{l=-\infty}^{\infty} P_l = 1$, where

$$P_l \equiv \sum_{n=0}^{\infty} |\langle n, l | \alpha \rangle|^2 = \frac{\mathcal{N}_\alpha}{\lambda^2} \Lambda_l \zeta(2, q_l), \tag{31}$$

is the occupation of the states with the given axial quantum number $l$ and

$$\Lambda_l = \left| \sum_{s'=1}^{s} \frac{V_{s'}}{V_0} e^{-2i\pi l \zeta_{s'}} \langle \mathbf{R}_{s'} | \alpha \rangle_{reg} \right|^2. \tag{32}$$

Similarly, for IPR we have

$$\eta \equiv \sum_{l=-\infty}^{\infty} \sum_{n=0}^{\infty} |\langle n, l | \alpha \rangle|^4 = \frac{\mathcal{N}_\alpha^2}{\lambda^4} \sum_{l=-\infty}^{\infty} \Lambda_l^2 \zeta(4, q_l). \tag{33}$$

The expressions above are used for T-noninvariant models (non-symmetric and symmetric), where $A \neq 0$ and $\langle \mathbf{R}_{s'} | \alpha \rangle_{reg}$ are complex. In the T-invariant models ($A = 0$) $\langle \mathbf{R}_{s'} | \alpha \rangle_{reg}$ are real. For PBC the normalization condition can be expressed as $\sum_{l=0}^{\infty} P_l^T = 1$, where

$$P_l^T = \frac{\mathcal{N}_\alpha}{\lambda^2} (2 - \delta_{l0})(\Lambda_l^c + \Lambda_l^s) \zeta(2, q_l), \tag{34}$$

and

$$\Lambda_l^{c,s} = \left( \sum_{s'=1}^{s} \frac{V_{s'}}{V_0} \langle \mathbf{R}_{s'} | \alpha \rangle_{reg} \left\{ \begin{array}{c} \cos 2\pi l \zeta_{s'} \\ \sin 2\pi l \zeta_{s'} \end{array} \right\} \right)^2. \tag{35}$$

Respectively, IPR can be expressed as

$$\eta = \frac{\mathcal{N}_\alpha^2}{\lambda^4} \sum_{l=0}^{\infty} (2 - \delta_{l0})(\Lambda_l^c + \Lambda_l^s)^2 \zeta(4, q_l). \tag{36}$$

For HWB we have the normalization condition $\sum_{l=1}^{\infty} P_l^B = 1$ with

$$P_l^B = \frac{\mathcal{N}_\alpha}{\lambda^2} \Lambda_l^B \zeta(2, q_l^B), \tag{37}$$

and

$$\eta = \frac{\mathcal{N}_\alpha^2}{\lambda^4} \sum_{l=1}^{\infty} \left( \Lambda_l^B \right)^2 \zeta(4, q_l^B), \tag{38}$$

where

$$\Lambda_l^B = \left( \sum_{s'=1}^{s} \frac{V_{s'}}{V_0} \langle \mathbf{R}_{s'} | \alpha \rangle_{reg} \sin \pi l \zeta_{s'} \right)^2, \tag{39}$$

and $q_l^B = (\pi^2 l^2 / 4 - \varepsilon_\alpha)/\lambda$.

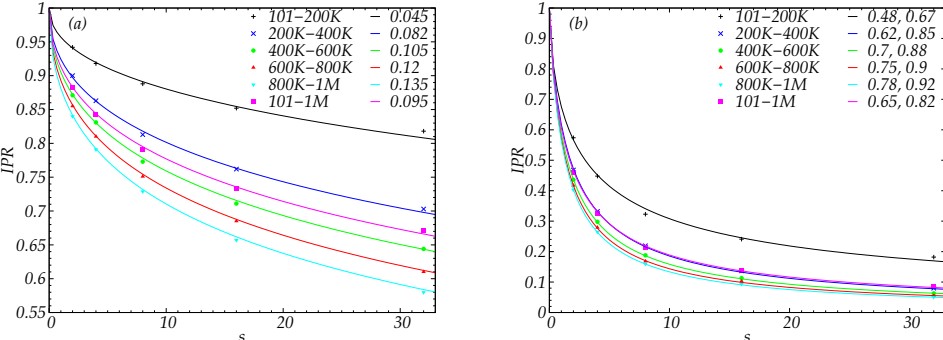

Figure 11: Inverse participation ratio as a function of the number of scatterers calculated for the non-symmetric model with $V_{s'}/V_0 = 10^{-3}$ (a) and $V_{s'}/V_0 = 10^{-2}$ (b) at various regions of the non-integrable system eigenstate labels $\alpha$. The lines show the dependencies $\eta_s = 1/(1 + v's^{0.48})$ with the $v'$ values presented in the legend in the part (a) and $\eta_s = 1/(1 + v's^\gamma)$ with the $v'$ and $\gamma$ values presented in the legend in the part (b).

A recurrence relation $\eta_s^{-1} = \eta_{s-1}^{-1} + v$ was derived [55] for NPC $\eta_s^{-1}$ of the system with $s$ scatterers. This means that NPC increases and, respectively, IPR decreases with the number of scatterers. In the case of weak interaction the dependence of NPC on the number of scatterers is nonlinear (see Fig. 11). This is a consequence of the strong dependence of the system's chaotic properties on the number of scatterers. NPC also increases with the eigenstate energy due to increase of the energy level density. In the case of the statistics of energy spectra, this increase was compensated by decrease of the effective interaction strength (see Fig. 5 and the related discussion above). Here we see that the wavefunction properties are determined by the interaction strength $V_{s'}$ rather than the effective one. The NPC dependence on the number of scatterers can be approximated by $\eta_s^{-1} = 1 + v's^\gamma$. For a weak interaction $V_{s'} = 10^{-3}V_0$ the power $\gamma \approx 0.48$ becomes independent of the eigenstate energy (see Fig. 11(a)). For stronger interaction $V_{s'}/V_0 = 10^{-2}$ (see Fig. 11(b)) the power $\gamma$ increases with the eigenstate energy

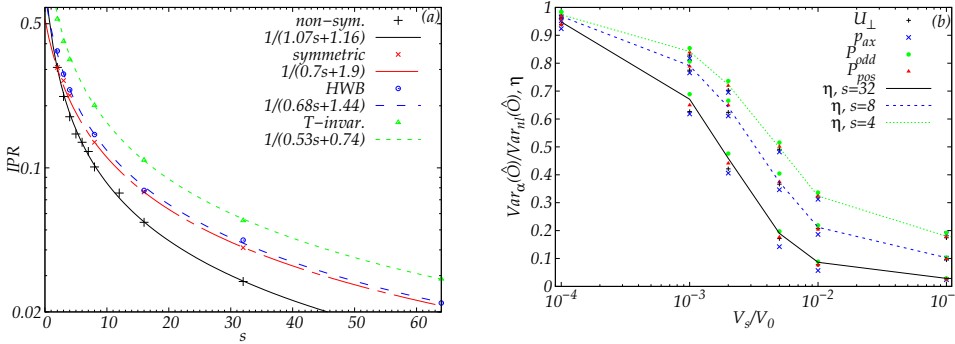

Figure 12: (a) Inverse participation ratio as a function of the number of scatterers calculated in the unitary regime for four kinds of the model. The lines show the best fit by inversely linear functions. This part uses the same data as Fig. 3(a) in [55]. (b) Ratio of fluctuation variances between the non-integrable and integrable systems eigenstates as a function of the scatterer strength for the non-symmetric model with 4, 8, and 32 scatterers. The points correspond to the four observables, the lines connect the calculated IPR values. The data for 32 scatterers are the same as in Fig. 3(b) of [55].

and the dependence of NPC on $s$ tends to the linear one. This means that $\nu$ is independent of $s$ since the system's chaotic properties are independent of the number of scatterers. In the unitary regime, this dependence is confirmed by IPR calculated for all kinds of the model (see Fig. 12(a)) which is approximated by inverse-linear functions with a good accuracy. We can see that for each number of scatterers the non-symmetric model has the minimal IPR and, therefore, demonstrates the highest chaoticity, the T-invariant PBC model has the highest IPR, and the HWB one lies between them. This order agrees with the NND and spectral rigidity of energy spectra for these models discussed in Sec. 3 above. However, the symmetric T-noninvariant model has substantially higher IPR than the non-symmetric one, although properties of energy spectra of these models demonstrate similar chaoticity. This difference can be related to properties of real and complex random Gaussian variables [55]. As well as any characteristic of chaos, IPR depends also on the interaction strength (see Fig. 12(b)). This figure also demonstrates that the systems with 4, 8, and 32 scatterers have approximately the same IPR ($\eta \approx 0.2$) at $V_{s'}/V_0 = 10^{-1}$ (and in the unitary regime), $10^{-2}$, and $5 \times 10^{-3}$, respectively, in agreement with the energy spectra statistics (see Fig. 8).

Chaotic properties of physical systems are also characterized by fluctuations of observable expectation values. Expectation value of the observable $\hat{O}$ in eigenstates of the non-integrable system is related to ones in integrable system eigenstates

$$\left\langle \alpha \left| \hat{O} \right| \alpha \right\rangle = \sum_{n,l,n',l'} \left\langle \alpha | n',l' \right\rangle \left\langle n',l' \left| \hat{O} \right| n,l \right\rangle \left\langle n,l | \alpha \right\rangle , \qquad (40)$$

where the expansion coefficients $\langle n,l|\alpha \rangle$ are given by (28).

Four observables are considered here. The transverse potential energy $m\omega_\perp^2 \rho^2/2$ is non-diagonal in the integrable system eigenstates

$$\left\langle n',l' \left| \frac{1}{2} m\omega_\perp^2 \rho^2 \right| n,l \right\rangle = \frac{\omega_\perp}{2} \delta_{ll'} \left[ (2n+1)\delta_{nn'} - n\delta_{n'n-1} - n'\delta_{nn'-1} \right] . \qquad (41)$$

As the potential energy increases with the total energy, the part $U_\perp$ of the transverse potential energy in the total energy is considered here. Its expectation value in the non-integrable system eigenstates can be expressed as (see Appendix D)

$$\langle \alpha | U_\perp | \alpha \rangle = \frac{\omega_\perp}{2E_\alpha} \left[ 1 + 2\frac{\mathcal{N}_\alpha}{\lambda^2} \sum_{l=-\infty}^{\infty} \Lambda_l \left( 1 - q_l \zeta(2,q_l) \right) \right] , \qquad (42)$$

for T-noninvariant models. In the T-invariant PBC case we have

$$\langle \alpha | U_\perp | \alpha \rangle = \frac{\omega_\perp}{2E_\alpha} \left[ 1 + 2\frac{\mathcal{N}_\alpha}{\lambda^2} \sum_{l=0}^{\infty} (2 - \delta_{l0})(\Lambda_l^c + \Lambda_l^s)\left( 1 - q_l \zeta(2,q_l) \right) \right] . \qquad (43)$$

In the last case, HWB, the expectation value takes the form

$$\langle \alpha | U_\perp | \alpha \rangle = \frac{\omega_\perp}{2E_\alpha} \left[ 1 + 2\frac{\mathcal{N}_\alpha}{\lambda^2} \sum_{l=1}^{\infty} \Lambda_l^B \left( 1 - q_l^B \zeta(2,q_l^B) \right) \right] . \qquad (44)$$

Other observables, diagonal in integrable system eigenstates, are the axial momentum $\langle nl |\hat{p}_{ax}| n'l' \rangle = l\delta_{n'n}\delta_{l'l}$, the occupation of positive momenta $\langle nl |\hat{P}_{pos}| n'l' \rangle = \delta_{n'n}\delta_{l'l}\theta(l)$, where $\theta(l) = 0$ for $l < 0$, $1/2$ for $l = 0$, and $1$ for $l > 0$, and the occupation of the odd axial modes $\langle nl |\hat{P}_{odd}| n'l' \rangle = \delta_{n'n}\delta_{l'l}\delta_{l\mathrm{mod}2,1}$, where $l\mathrm{mod}2$ is the reminder of the division of $l$ by 2. For T-noninvariant models their expectation values are expressed in terms of the occupations $P_l$ (31),

$$\langle \alpha |\hat{p}_{ax}| \alpha \rangle = \sum_{l=-\infty}^{\infty} l P_l , \quad \left\langle \alpha \left| \hat{P}_{pos} \right| \alpha \right\rangle = \frac{1}{2} P_0 + \sum_{l=1}^{\infty} P_l , \quad \left\langle \alpha \left| \hat{P}_{odd} \right| \alpha \right\rangle = \sum_{l=-\infty}^{\infty} P_{2l+1} . \qquad (45)$$

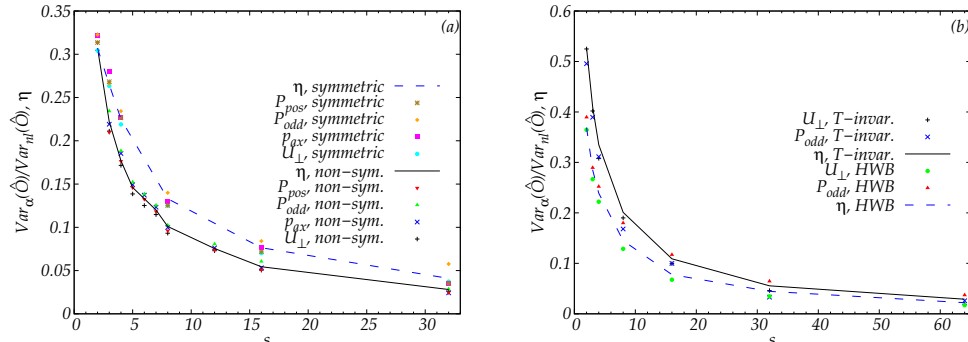

Figure 13: Ratio of fluctuation variances between the non-integrable and integrable systems eigenstates as a function of the number of scatterers for T-noninvariant (a) and T-invariant (b) models in the unitary regime. The points correspond to the four observables, the lines connect the calculated IPR values. The data for the non-symmetric model in the part (a) are the same as in Fig. 3(c) of [55].

For T-invariant models, $\langle \alpha | \hat{p}_{ax} | \alpha \rangle = 0$ and $\langle \alpha | \hat{P}_{pos} | \alpha \rangle = 1/2$ do not fluctuate, while $\langle \alpha | \hat{P}_{odd} | \alpha \rangle = \sum_{l=1}^{\infty} P_{2l-1}^{T,B}$ are expressed in terms of probabilities (34) and (37), respectively.

For an observable $\hat{O}$, the variance of its expectation value fluctuations between non-integrable system eigenstates is defined as

$$\text{Var}_\alpha(\hat{O}) = \overline{\langle \alpha | \hat{O} | \alpha \rangle^2} - \overline{\langle \alpha | \hat{O} | \alpha \rangle}^2. \tag{46}$$

According to [32], this variance is proportional to IPR and the variance between the integrable system eigenstates

$$\text{Var}_\alpha(\hat{O}) = \eta \text{Var}_{nl}(\hat{O}). \tag{47}$$

The latter variances are calculated in Appendix for the four observables presented above. The variance of the axial momentum

$$\text{Var}_{nl}(p_{ax}) = (\varepsilon_{max}^{5/2} - \varepsilon_{min}^{5/2})/[5\pi^2(\varepsilon_{max}^{3/2} - \varepsilon_{min}^{3/2})], \tag{48}$$

depends on the averaging interval $[\varepsilon_{min}, \varepsilon_{max}]$ boundaries. The variances of other observables are independent of the interval, $\text{Var}_{nl}(\hat{P}_{pos}) = 1/4$, $\text{Var}_{nl}(\hat{P}_{odd}) = 1/4$, and $\text{Var}_{nl}(\hat{U}_\perp) = 1/45$. Figure 12(b) confirms the rule (47) for the integrability-chaos transition on variation of the scatterer strength in the non-symmetric model, both for 4, 8, and 32 scatterers. This rule is also confirmed when the number of scatterers is changed for all four models considered here (see Fig. (13)).

# 5 Conclusion

An effective method of numerical solution, based on properties of high-rank separable perturbations, is developed for a harmonic waveguide with a vector potential and either PBC or HWB in the axial direction, perturbed by zero-range scatterers along the waveguide axis. The energy-degeneracy of the unperturbed system can be lifted by the vector potential which also lifts T-invariance. The energy spectra properties — near-neighbor distribution and spectral rigidity, as well as IPR and fluctuation variance of observable expectation values, are calculated for $10^6$ eigenstates. The chaoticity measures of the model increase with the number of scatterers and their strengths. This allows exploring the integrability-chaos transition.

In T-noninvariant models, the energy spectra properties follow the Šeba plots already for 2 scatterers and approach the Wigner-Dyson predictions for 32 scatterers. The model with non-symmetric scatterer distribution approaches the GUE statistics, while the P-invariant distribution leads to the GOE statistics inherent in T-invariant systems. It is a consequence of PT-invariance of the latter model, leading to a real interaction matrix. Similarly, the IPR difference between the two kinds of models can be related to properties of real and complex wavefunctions.

The T-invariant HWB and PBC models approach the Šeba statistics only for 16 and 32 scatterers, respectively, and the GOE one for 32 and 64 scatterers, respectively, i.e., much slower than the T-noninvariant models. This can be related to the vector potential, which randomizes the sequence of quantum numbers of energy-ordered eigenstates in the integrable system.

Calculation for different numbers of scatterers and their strengths confirm the prediction [55] that IPR decreases with the number of scatterers. The dependence is inversely proportional for strong scatterers. The prediction [32] that the ratio of the observable fluctuation variances for the nonintegrable and integrable systems is approximately equal to IPR is confirmed as well. Thus, all criteria of chaoticity confirm that the model approaches the complete quantum chaos and the eigenstate thermalization when the number of scatterers is increased.

# A Derivation of the summands $T_n(\zeta_{s'}, \zeta_{s''})$ and $T_n^{reg}$ in Eq. (15)

Let us define

$$T_n(\frac{z}{L}, \frac{z'}{L}) = \frac{2\pi a_\perp^2}{mL} \sum_l \frac{\langle 0,0,z|nl\rangle \langle nl|0,0,z'\rangle}{E - E_{nl}}\,. \tag{A.1}$$

For the PBC models using Eqs. (4), (5), and (7) we get

$$T_n(\zeta, \zeta') = \sum_{l=-\infty}^{\infty} \frac{\exp(2i\pi l(\zeta - \zeta'))}{\varepsilon - \lambda n - \pi^2(l - l_0)^2}\,, \tag{A.2}$$

where $\varepsilon = mL^2(E - \omega_\perp)/2$. Due to translational invariance of PBC, $T_n$ is a function of $z - z'$ only. Then $T_n(\zeta_{s'}, \zeta_{s''}) = T_n(\zeta_{s'} - \zeta_{s''}, 0)$, and, therefore, only $T_n(\zeta, 0)$ should be evaluated. Farther, the partial fraction decomposition

$$\frac{1}{\varepsilon - \lambda n - \pi^2(l - l_0)^2} = \frac{1}{2\pi p_n}\left(\frac{1}{l - l_0 + p_n/\pi} - \frac{1}{l - l_0 - p_n/\pi}\right), \tag{A.3}$$

where $p_n = \sqrt{\varepsilon - \lambda n}$, allows us to use the summation formula

$$\sum_{l=-\infty}^{\infty} \frac{\exp(2i\pi l\zeta)}{l + a} = \frac{\pi}{\sin\pi a}\exp(-2i\pi a(\zeta - [\zeta] - 1/2)), \tag{A.4}$$

following from Eq. (5.4.3.4) in [68]. As $0 \le \zeta < 1$, this leads to

$$T_n(\zeta, 0) = \frac{1}{2p_n}e^{2i\pi l_0\zeta}\left[e^{2ip_n\zeta}(\cot(\pi l_0 + p_n) - i) - e^{-2ip_n\zeta}(\cot(\pi l_0 - p_n) - i)\right]. \tag{A.5}$$

In the diagonal elements of the matrix $S_{s's''}(\varepsilon)$ [see Eq. (15)] we need

$$T_n(0,0) = \frac{\sin 2p_n}{p_n(\cos 2\pi l_0 - \cos 2p_n)}\,. \tag{A.6}$$

When $\lambda n > \varepsilon_\alpha$, $p_n$ becomes imaginary, $|p_n| = \sqrt{\lambda n - \varepsilon_\alpha}$, and we have

$$T_n(\zeta,0) = -\frac{1}{|p_n|}e^{2i\pi l_0\zeta}\left(\frac{e^{-2|p_n|\zeta}}{1-e^{2i\pi l_0-2|p_n|}} + \frac{e^{-2|p_n|(1-\zeta)}}{e^{2i\pi l_0}-e^{-2|p_n|}}\right). \tag{A.7}$$

In the limit of the large $n$ and for any $0 < \zeta < 1$ the two terms in the parentheses decay as $\exp(-2\sqrt{\lambda n}\zeta)$ and $\exp(-2\sqrt{\lambda n}(1-\zeta))$, respectively. However, if $\zeta = 0$, $T_n(0,0) \sim n^{-1/2}$ and the sum of $T_n(0,0)$ diverges. In order to regularize this sum, let us represent $T_n(\zeta,0)$ in the limit of $\zeta \to 0$ as

$$T_n(\zeta,0) \sim -\frac{e^{-2|p_n|\zeta}}{|p_n|} + T_n^{reg}, \quad T_n^{reg} = -\frac{2}{|p_n|}\frac{(\cos 2\pi l_0 - e^{-2|p_n|})e^{-2|p_n|}}{(e^{-2|p_n|}-2\cos 2\pi l_0)e^{-2|p_n|}+1}. \tag{A.8}$$

$T_n^{reg}$ decreases exponentially with $n$ and, due to the translational invariance, it is independent of $\zeta$. In the limit of $\zeta \to 0$, the sum of the first terms in $T_n(\zeta,0)$ was calculated in [62]

$$\sum_{n=n_0}^{\infty}\frac{e^{-2|p_n|\zeta}}{|p_n|} \sim \frac{1}{\lambda\zeta} + \frac{1}{\sqrt{\lambda}}\zeta\left(\frac{1}{2}, n_0 - \frac{\varepsilon}{\lambda}\right), \tag{A.9}$$

in terms of the Hurwitz zeta function (see [61]). The first, proportional to $\zeta^{-1}$, term here is removed by the derivative in (13). Then we get Eqs. (14) and (15).

For T-invariant models, when $A = 0$, we have real $T_n(\zeta,0)$. In the case of PBC, we can just set $l_0 = 0$ in Eqs. (A.5), (A.7), and (A.8) and get

$$\begin{aligned}
T_n(\zeta,0) &= \frac{\cos p_n(1-2\zeta)}{p_n\sin p_n} \quad (\lambda n < \varepsilon), \\
T_n(\zeta,0) &= -\frac{1}{|p_n|}\frac{e^{-2|p_n|\zeta}+e^{-2|p_n|(1-\zeta)}}{1-e^{-2|p_n|}} \quad (\lambda n > \varepsilon, \zeta > 0), \\
T_n^{reg} &= -\frac{2}{|p_n|}\frac{e^{-2|p_n|}}{1-e^{-2|p_n|}}.
\end{aligned} \tag{A.10}$$

In the case of HWB, substitution of Eqs. (6) and (9) to (A.1) leads to

$$T_n(\zeta,\zeta') = \sum_{l=1}^{\infty}\frac{\cos(\pi l(\zeta-\zeta'))-\cos(\pi l(\zeta+\zeta'))}{\varepsilon-\lambda n-\pi^2 l^2/4}. \tag{A.11}$$

Unlike (A.2), it is not a function of $z-z'$ only, since HWB is not translational invariant. Using partial fraction decomposition and the real part of the summation formula (A.4), we get for $\zeta > \zeta'$

$$T_n(\zeta,\zeta') = -2\frac{\sin 2p_n(1-\zeta)\sin 2p_n\zeta'}{p_n\sin 2p_n}. \tag{A.12}$$

For $\lambda n > \varepsilon_\alpha$ and $\zeta > \zeta'$ we have

$$T_n(\zeta,\zeta') = -\frac{1}{|p_n|}\frac{e^{-2|p_n|(2-\zeta+\zeta')}+e^{-2|p_n|(\zeta-\zeta')}-e^{-2|p_n|(2-\zeta-\zeta')}-e^{-2|p_n|(\zeta+\zeta')}}{1-e^{-4|p_n|}}. \tag{A.13}$$

The term causing the divergence is separated in the same way as in Eq. (A.8), providing

$$T_n^{reg}(\zeta) = -\frac{1}{|p_n|}\frac{2e^{-4|p_n|}-e^{-4|p_n|\zeta}-e^{-4|p_n|(1-\zeta)}}{1-e^{-4|p_n|}}. \tag{A.14}$$

# B Eigenvalues of the system (14) matrix

Let us arrange the eigenenergies of the integrable system in increasing order and label them by an index $k$ such that $\varepsilon_k \equiv \varepsilon_{n_k l_k}$ and $\varepsilon_k < \varepsilon_{k+1}$. The term $T_{n_k}(\zeta, \zeta')$ has a singularity as a function of $\varepsilon$ when $\varepsilon \to \varepsilon_k$ and can be separated to singular and continuous parts, $T_{n_k}(\zeta, \zeta') = T_k^{sing}(\zeta, \zeta') + T_k^{cont}(\zeta, \zeta')$. For PBC, $p_{n_k} \sim \pi|l_k - l_0| + (\varepsilon - \varepsilon_k)/(2\pi|l_k - l_0|)$ in the limit $\varepsilon \to \varepsilon_k$ and these parts are expressed as

$$T_k^{sing}(\zeta, \zeta') = \frac{1}{2p_{n_k}\sin(p_{n_k} - \pi|l_k - l_0|)}\exp\left(2i(\pi l_0 + \tilde{p}_k)(\zeta - \zeta')\right),$$

$$T_k^{cont}(\zeta, 0) = -\frac{1}{2\tilde{p}_k}e^{2i\pi l_0\zeta}\left[e^{2i\tilde{p}_k\zeta}\left(\tan\frac{\pi(l_k - l_0) + \tilde{p}_k}{2} + i\right) + e^{-2i\tilde{p}_k\zeta}\left(\cot(\pi l_0 - \tilde{p}_k) - i\right)\right],$$
(B.1)

where $\tilde{p}_k = p_{n_k}\text{sign}(l_k - l_0)$. In the T-invariant case, when $l_k \neq 0$, they can be expressed as

$$T_k^{sing}(\zeta, \zeta') = \frac{\cot p_{n_k}}{p_{n_k}}\left(\cos 2p_{n_k}\zeta \cos 2p_{n_k}\zeta' + \sin 2p_{n_k}\zeta \sin 2p_{n_k}\zeta'\right),$$

$$T_k^{cont}(\zeta, \zeta') = \frac{\sin 2p_{n_k}(\zeta - \zeta')}{p_{n_k}}.$$
(B.2)

If $l_k = 0$, $p_{n_k} = \sqrt{\varepsilon - \varepsilon_k}$, and the second term in the parenthesis in $T_k^{sing}$ becomes non-singular and is moved to $T_k^{cont}$.

For HWB, when $l_k \neq 0$, we have $p_{n_k} \sim \pi l_k/2 + (\varepsilon - \varepsilon_k)/(\pi l_k)$ and

$$T_k^{sing}(\zeta, \zeta') = 2\frac{\cot 2p_{n_k}}{p_{n_k}}\sin 2p_{n_k}\zeta \sin 2p_{n_k}\zeta',$$

$$T_k^{cont}(\zeta, \zeta') = -2\frac{\cos 2p_{n_k}\zeta \sin 2p_{n_k}\zeta'}{p_{n_k}}.$$
(B.3)

If $l_k = 0$, $T_{n_k}(\zeta, \zeta')$ is non-singular.

In any case, for $(\varepsilon_{k-1} + \varepsilon_k)/2 < \varepsilon < (\varepsilon_k + \varepsilon_{k+1})/2$ the matrix $S_{s's''}(\varepsilon)$ (15) can be represented as $S_{s's''}(\varepsilon) = T_k^{sing}(\zeta_{s'}, \zeta_{s''}) + S_{s's''}^{cont}(\varepsilon)$, where $S_{s's''}^{cont}(\varepsilon)$ is continuous. The singular part can be expressed in terms of orthonormal vectors $b_i(\zeta_{s'})$

$$T_k^{sing}(\zeta_{s'}, \zeta_{s''}) = \sum_{i=1}^{i_{max}} B_i b_i^*(\zeta_{s'})b_i(\zeta_{s''}), \quad \sum_{s'=1}^{s} b_{i'}^*(\zeta_{s'})b_i(\zeta_{s'}) = \delta_{ii'},$$
(B.4)

and has a form of the matrix with $i_{max}$ eigenvalues $B_i$. When $\varepsilon$ approaches $\varepsilon_k$, the singular part dominates and the eigenvalues tend to $\pm\infty$. Then in the T-invariant PBC case with $l_k \neq 0$ we have $i_{max} = 2$ and two eigenvalues of the matrix $S_{s's''}(\varepsilon)$ have singularities at $\varepsilon \to \varepsilon_k$, there are no singular eigenvalues ($i_{max} = 0$) in the case of HWB with $l_k = 0$, and single eigenvalue has a singularity in other cases when $i_{max} = 1$. Results of numerical calculations in Fig. 14 demonstrate these properties. They also show that the eigenvalues decrease monotonically with $\varepsilon$. Then each eigenvalue can have single root in the interval $[\varepsilon_k, \varepsilon_{k+1}]$. In Fig. 14, the number of eigenvalues with roots increases from 0 to 4 in parts (a)-(e).

In the close vicinity of $\varepsilon_k$ direct numerical diagonalization of the matrix $S_{s's''}(\varepsilon)$ becomes inaccurate if $i_{max} > 0$. However, in this vicinity $i_{max}$ eigenvalues are approximated by $B_i$ with good accuracy. In order to calculate other eigenvalues, the matrix $S_{s's''}^{cont}(\varepsilon)$ is projected out of the envelope of the vectors $b_i(\zeta_{s'})$,

$$\sum_{s'',s'''}\left(\delta_{s's''} - \sum_{i=1}^{i_{max}} b_i^*(\zeta_{s'})b_i(\zeta_{s''})\right)S_{s''s'''}^{cont}(\varepsilon)\left(\delta_{s'''s^{iv}} - \sum_{i=1}^{i_{max}} b_i^*(\zeta_{s'''})b_i(\zeta_{s^{iv}})\right).$$
(B.5)

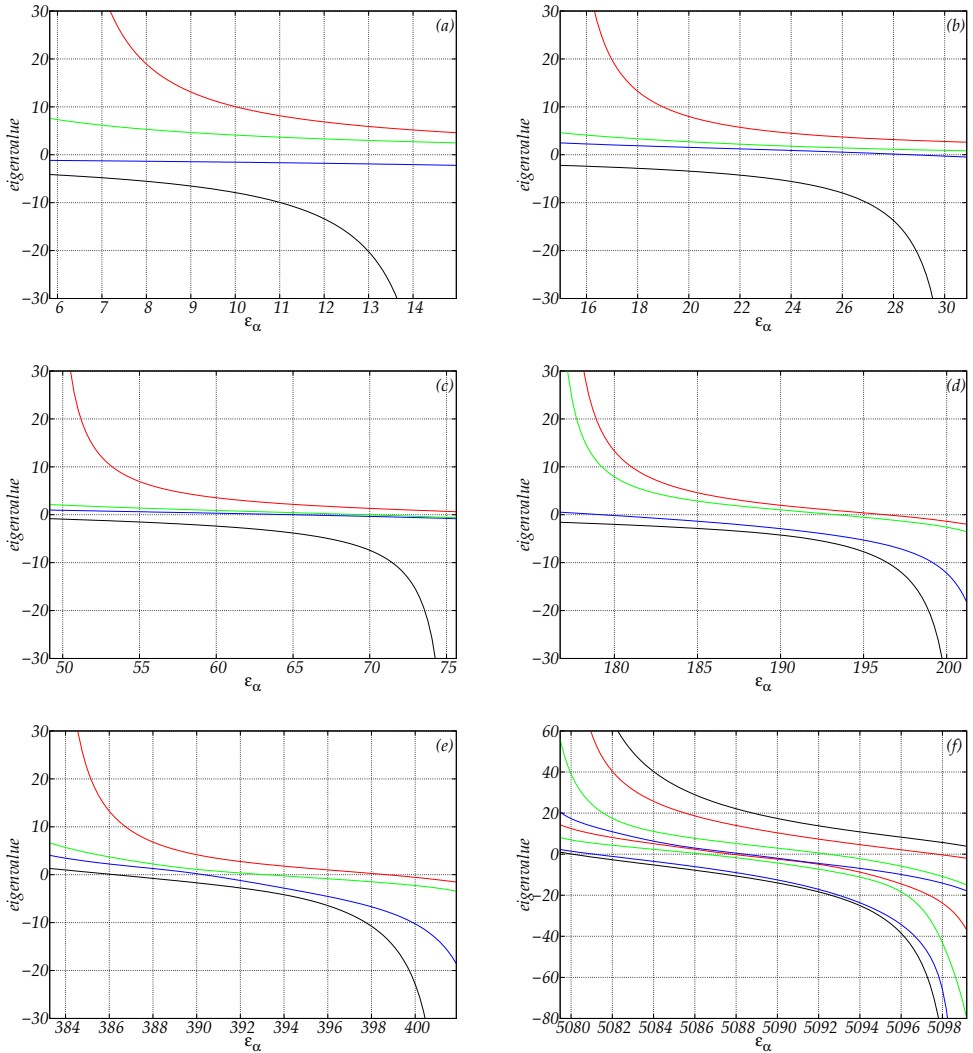

Figure 14: Examples of eigenvalue dependence on the energy between two neighboring eigenenergies of the integrable system for the non-symmetric model with 4 scatterers (a-e) and the T-invariant model with 8 scatterers (f).

Numerical diagonalization of this matrix provides $i_{max}$ eigenvalues which are close to zero (they correspond to eigenvectors $b_i(\zeta_{s'})$), other eigenvalues approximate the remained $s - i_{max}$ eigenvalues of $S_{s's''}(\varepsilon)$.

## C Level spacing ratio

The ratio of two consecutive level spacings [66, 67]

$$r_\alpha = \frac{\min(E_{\alpha+1} - E_\alpha, E_\alpha - E_{\alpha-1})}{\max(E_{\alpha+1} - E_\alpha, E_\alpha - E_{\alpha-1})}, \tag{C.1}$$

can characterize the energy spectrum statistics and does not require unfolding. Its averages $\langle r \rangle$ were calculated in [67] for the Poisson ($\langle r \rangle = 2\ln 2 - 1 \approx 0.38629$), GOE ($\langle r \rangle = 4 - 2\sqrt{3} \approx 0.53590$), and GUE ($\langle r \rangle = 2\sqrt{3}/\pi - 1/2 \approx 0.60266$) statistics. Figure 15(a) shows that for the present model $\langle r \rangle$ increases at weak interactions, but demonstrate non-monotonic dependence when the value $\langle r \rangle \approx 6$, corresponding to GUE, is approached. In some

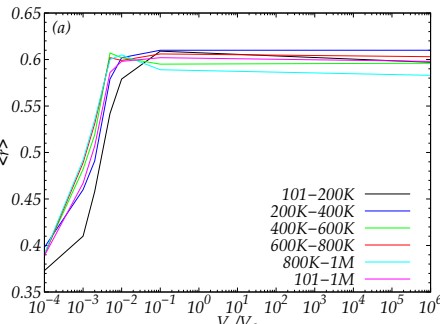
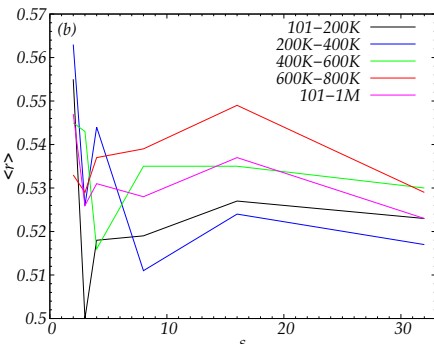

Figure 15: The level spacing ratio averaged over different eigenstate label intervals (a) for the non-symmetric model with 32 scatterers as a function of the interaction strength and (b) for the symmetric model in the unitary regime as a function of the number of scatteres.

eigenstate intervals, the level spacing ratio has maximum $\langle r \rangle \approx 6$ already at $V = 5 \times 10^{-3} V_0$, in contradiction with NND and spectral rigidity (cf. Fig. 7). When the number of scatterers is increased (see Fig. 15(b)), $\langle r \rangle$ non-monotonically decreases, although the monotonic increase of chaoticity is demonstrated by the NND change from Šeba to GOE predictions, as well as by the spectral rigidity (see Fig. 2).

# D  Expectation values

Substituting Eqs. (28) and (41) into Eq. (40) we can get the following expression for the expectation value of the transverse potential energy in the non-integrable system eigenstates

$$
\begin{aligned}
\left\langle \alpha \left| \frac{1}{2} m \omega_\perp^2 \rho^2 \right| \alpha \right\rangle &= \frac{\omega_\perp}{2} \mathcal{N}_\alpha \sum_{l=-\infty}^{\infty} \Lambda_l \sum_{n,n'} \frac{(2n+1)\delta_{nn'} - n\delta_{n'n-1} - n'\delta_{nn'-1}}{(\varepsilon_\alpha - \varepsilon_{nl})(\varepsilon_\alpha - \varepsilon_{n'l})} \\
&= \frac{\omega_\perp}{2} \left\{ 1 + 2\frac{\mathcal{N}_\alpha}{\lambda^2} \sum_{l=-\infty}^{\infty} \Lambda_l \sum_{n=0}^{\infty} \left[ \frac{n}{(q_l+n)^2} - \frac{n}{(q_l+n)(q_l+n-1)} \right] \right\}, \quad \text{(D.1)}
\end{aligned}
$$

where the last transformation uses the normalization condition, Eq. (7) for $\varepsilon_{nl}$, and Eq. (30) for $q_i$. The sum over $n$ here can be transformed as

$$
\sum_{n=0}^{\infty} \left[ -\frac{q_l}{(q_l+n)^2} + \frac{q_l-1}{(q_l+n)(q_l+n-1)} \right] = -q_l\zeta(2,q_l) + (q_l-1)\sum_{n=0}^{\infty}\left( \frac{1}{q_l+n-1} - \frac{1}{q_l+n} \right),
$$
(D.2)

where the summation over $n$ with Eq. (29) for the first term in the square brackets and partial fraction decomposition for the second term are used. The last sum over $n$ is reduced to $1/(q_l-1)$ due to cancellation of the terms. This leads to Eq. (42). The derivation above is related to the T-noninvariant models. The same transformation of the sum over $n$ leads to the expectation values for the T-invariant PBC (43) and HWB (44) models.

The variance between the integrable system eigenstates can be evaluated analytically. The product $\bar{\alpha}(\varepsilon)\overline{U_\perp}$ can be approximated by the sum

$$
\sum_{n,l} \langle nl | \hat{U}_\perp | nl \rangle \theta(\varepsilon - \varepsilon_{nl}) = \frac{\lambda}{2} \sum_{n=0}^{[\varepsilon/\lambda]} \left(n + \frac{1}{2}\right) \sum_{l=l_{min}(n)}^{l_{max}(n)} \frac{1}{\varepsilon_{nl} + \lambda/2}, \quad \text{(D.3)}
$$

where $l_{min,max}(n) = l_0 \mp \sqrt{\varepsilon - \lambda n}/\pi$ and Eqs. (7) and (41) are used. Replacing summation by integration and neglecting the values $\sim 1$ compared to $n$, we approximate the sum as

$$\frac{\lambda}{2} \int_0^{\varepsilon/\lambda} n\, dn \int_{l_{min}}^{l_{max}} dl\, \frac{1}{\varepsilon_{nl}} = \frac{4}{9\pi\lambda} \varepsilon^{3/2}. \tag{D.4}$$

It has the same $\varepsilon$ dependence as $\bar{\alpha}(\varepsilon)$ taken with the same accuracy [the first term in Eq. (19)]. Then the average $\overline{U_\perp} = 1/3$ is independent of the averaging interval (this value agrees to the virial theorem). In the same way we find $\overline{U_\perp^2} = 2/15$ and, therefore, $\mathrm{Var}_{nl}(\hat{U}_\perp) = 1/45$. Although in the HWB model $l \geq 0$, we get the same results due to the distinction between Eqs. (7) and (9).

For the average axial momentum, we approximately evaluate the sum

$$\sum_{n,l} l\,\theta(\varepsilon - \varepsilon_{nl}) \approx \int_{l_{min}(0)}^{l_{max}(0)} l\, dl \int_0^{n_{max}} dn = \frac{4}{3\pi\lambda} \varepsilon^{3/2} l_0, \tag{D.5}$$

where $n_{max} = [\varepsilon - \pi^2(l - l_0)^2]/\lambda$. This leads to $\overline{p_{ax}} = l_0$. However, evaluating $\overline{p_{ax}^2}$, we see that

$$\sum_{n,l} l^2 \theta(\varepsilon - \varepsilon_{nl}) \approx \frac{4}{3\pi\lambda} \left( \varepsilon^{3/2} l_0^2 + \frac{1}{5\pi^2} \varepsilon^{5/2} \right), \tag{D.6}$$

has a different $\varepsilon$ dependence. Therefore,

$$\overline{p_{ax}^2} = l_0^2 + \frac{1}{5\pi^2} \frac{\varepsilon_{max}^{5/2} - \varepsilon_{min}^{5/2}}{\varepsilon_{max}^{3/2} - \varepsilon_{min}^{3/2}}, \tag{D.7}$$

depends on the averaging interval $[\varepsilon_{min}, \varepsilon_{max}]$ boundaries. As a result, we get the variance (48).

In the integrable system basis, $\overline{P_{pos}} = \overline{P_{odd}} = \overline{P_{pos}^2} = \overline{P_{odd}^2} = 1/2$. This leads to $\mathrm{Var}_{nl}(\hat{P}_{pos}) = \mathrm{Var}_{nl}(\hat{P}_{odd}) = 1/4$.

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
