# Peer review of "Quantum chaos in a harmonic waveguide with scatterers"

_SciPost Physics, doi:SciPost Phys. 15, 221 (2023)_

## Round 1 · Referee Report · Anonymous (Referee 1) · 2023-7-24

Strengths

1) The work addresses a fundamentally interesting and timely research question. Namely, the nature of the transition from integrability to chaos for a particle confined in an external potential – here a harmonic waveguide – in the presence of scatterers. With relevance to other integrable systems that undergo perturbations.
2) Numerical results presented in Sec. 3 contain genuinely new insight into this problem, and are neatly put into context of the existing literature.
3) The manuscript is well formatted, including a comprehensive appendix containing sufficient detail that the presented results could be reproduced by an expert reader.

Weaknesses

1) It is currently unclear whether aspects of the numerical results presented in Sec. 4 have already appeared in a previous work by the same author. If so, this should be clearly indicated in the text, and clear reasons outlining the significance of the new results presented in this work should be emphasised.
2) The motivation for why the author chose to investigate the four particular models discussed in the text is somewhat unclear. Is there some physical intuition behind this choice?
3) Grammatical structure and spelling of technical terms in the introduction could be improved.

Report

In this work, the author reports on a numerical study of quantum chaos in a harmonic waveguide with zero-range scatterers placed along its axis. Specifically, they investigate the transition between chaos and integrability in four different models (i.e. corresponding to different boundary conditions, vector potentials and positions of the scatterers) and study the fluctuations of physical observables as well as spectral statistics in each case. This is done by varying either the number or strength of the scatterers, as well as the strength of the scaled vector potential.

The paper is well formatted, and as best I can judge, the presented results are all technically correct. Moreover, the main text and appendices provide a sufficient level of mathematical detail so that an expert reader would be able to follow the author’s derivations, reasoning, and subsequent analysis without issue. The author’s study of the energy level statistics and spectral rigidity for this system contains genuinely new material which represents a valuable addition to the existing literature. However, I still have some questions regarding the numerical results presented in Sec. 4 of the manuscript.

The author’s paper reads a bit as if the numerical results and analysis presented in Sec. 4 are all completely new, however in a previous work (Yurovsky, PRL 130 (2023)) the same author investigates an identical model and observables. This previous work contains identical analysis of the linear relationship between the number of principal components (NPC) and the number of scatterers (see Fig. 3(a)), as is presented in Fig. 10 of the current manuscript. This is highlighted by the identical coefficients extracted from the fitting functions in both cases. Similar overlap also appears between Figs. 3(b)-(c) of this previous work, and Figs. 11-12(a) of the current manuscript. To remedy the situation, I suggest that the author appropriately reference data already published in this previous work, as well as comment on the key features distinguishing the results presented in the current manuscript (see Requested changes).

If the author can adequately address the comments raised here, then I believe that this manuscript would be suitable for publication in SciPost Physics.

Requested changes

Here are a few key points that the author should consider:

1) The similarity of the numerical results presented in this work and Fig. 3 from Ref. [54] (i.e. Yurovsky, PRL 130 (2023)) should be addressed. In particular, although the author references [54] in regard to the derivation of the recurrence relation $\eta_s^{-1}= \eta_{s-1}^{-1}+\nu$, nowhere is it mentioned that the numerical data presented in Figs. 10(a), Fig. 11 and Fig. 12(a) were already published (either in part or in full) in a prior work. In Sec. 4, it should be made clear exactly which data (if any) is shared between plots in the current work and Ref. [54], as well as outlining the significance of the new results presented solely in the current manuscript.

2) In Sec. 3, when analysing the transition from integrability to chaos using the nearest-neighbour distribution of level-spacings, why wasn’t the distribution of level spacing ratios used instead? Thereby avoiding the rather involved procedure of unfolding the spectrum? To my knowledge, using the distribution of level ratios is now standard practice when analysing the level-spacing statistics of quantum many-body systems (see e.g. Atas et al., PRL 110 (2013), Oganesyan et al., PRB 75 (2007)). As a follow up, it would also be preferable to calculate the mean value of the level-spacing distributions $\langle s \rangle$ or $\langle r \rangle$ , as a complementary method for quantifying the agreement between the numerical data and the GOE/Poissonian/Šeba distributions. A comparison of these mean values against the corresponding analytical prediction for the relevant ensemble would be much more robust than simply relying on visual agreement between histograms.

3) The overall grammatical structure and spelling of the manuscript could be improved. In particular, a technical term “Wigner-Dyson” (rather than Wigner-Dayson as used in the text) is consistently misspelled throughout the manuscript. Moreover, the inconsistent spelling of chaoticity/chaotisity in the introduction should be addressed.

4) Is it possible to calculate numerical results for $s > 64$ scatterers? Why were results for $s = 64$ scatterers shown only in Figs. 3-4, but not in Figs. 1-2?

5) Does there exist an analytical prediction for the spectral rigidity function corresponding to Šeba statistics? As expressed at the bottom of page 6 in the case of Poissonian, GOE or GUE statistics?

6) Is it possible to realise the Hamiltonian (Eq. (2)) under consideration in experiment? How would one tune the number, position and strength of the scatters, or alter the vector potential/boundary conditions in this case, in order to interpolate between the four specific models that are analysed? Would it be possible to probe the fluctuations in the specific observables considered in Sec. 4?

And a couple of more minor points:

7) There are some inconsistencies regarding the way equations are labelled in both the main text and appendices. For example, following Eq. (18) several equations displayed in the main text remain unlabelled, until Eq. (19) appears are the beginning of Sec. 4. To improve readability, I believe it would be best if all equations appearing in the main text/appendix were labelled in sequence, even those that are not directly referenced in the text.

8) In Fig. 7(a), multiple results are plotted with thin black lines. This makes it very difficult to distinguish between these data sets. In general, for plots of the level-spacing distributions presented in Sec. 3, it is often quite difficult to distinguish visually between the different legend entries, and then match these to the appropriate histogram. Perhaps employing some alternate line styles etc. could improve the clarity of these figures.

9) In Fig. 8(a), the third legend entry reads “GOI” rather than GOE.

  • validity: good
  • significance: good
  • originality: ok
  • clarity: ok
  • formatting: good
  • grammar: reasonable

Author:  Vladimir Yurovsky  on 2023-09-08  [id 3963]

(in reply to Report 1 on 2023-07-24)

I am indeed grateful to the Referee for careful reading of the manuscript and pointing out unclear statements. All Referee’s remarks are addressed. The revised manuscript contains necessary clarifications, as well as other corrections in response to the Referee's remarks. Following below are replies to all remarks of the Referee.

Weaknesses

1) It is currently unclear whether aspects of the numerical results presented in Sec. 4 have already appeared in a previous work by the same author. If so, this should be clearly indicated in the text, and clear reasons outlining the significance of the new results presented in this work should be emphasised.

This issue is clarified (see the response to the requested change 1 below).

2) The motivation for why the author chose to investigate the four particular models discussed in the text is somewhat unclear. Is there some physical intuition behind this choice?

It is now explained in the sentence inserted below the paragraph containing Eq. (9).

3) Grammatical structure and spelling of technical terms in the introduction could be improved.

Corrected.

Requested changes

Here are a few key points that the author should consider:

1) The similarity of the numerical results presented in this work and Fig. 3 from Ref. [54] (i.e. Yurovsky, PRL 130 (2023)) should be addressed. In particular, although the author references [54] in regard to the derivation of the recurrence relation η−1s=η−1s−1+ν , nowhere is it mentioned that the numerical data presented in Figs. 10(a), Fig. 11 and Fig. 12(a) were already published (either in part or in full) in a prior work. In Sec. 4, it should be made clear exactly which data (if any) is shared between plots in the current work and Ref. [54], as well as outlining the significance of the new results presented solely in the current manuscript.

I am indeed grateful to the referee for pointing out my omission. Captions to figures 12 and 13 (former 11 and 12) in the revised manuscript specify which data are shared. An additional subfigure (Fig. 11(b)) is included and Figs. 11 and 12 (former 10 and 11) are rearranged. The discussion of the case of weak interactions is extended and outlines significance of these results.

2) In Sec. 3, when analysing the transition from integrability to chaos using the nearest-neighbour distribution of level-spacings, why wasn’t the distribution of level spacing ratios used instead? Thereby avoiding the rather involved procedure of unfolding the spectrum? To my knowledge, using the distribution of level ratios is now standard practice when analysing the level-spacing statistics of quantum many-body systems (see e.g. Atas et al., PRL 110 (2013), Oganesyan et al., PRB 75 (2007)). As a follow up, it would also be preferable to calculate the mean value of the level-spacing distributions ⟨s⟩ or ⟨r⟩ , as a complementary method for quantifying the agreement between the numerical data and the GOE/Poissonian/Šeba distributions. A comparison of these mean values against the corresponding analytical prediction for the relevant ensemble would be much more robust than simply relying on visual agreement between histograms.

The level spacing ratio is now discussed in the last paragraph of Sec. 3 and in the new appendix C.

3) The overall grammatical structure and spelling of the manuscript could be improved. In particular, a technical term “Wigner-Dyson” (rather than Wigner-Dayson as used in the text) is consistently misspelled throughout the manuscript. Moreover, the inconsistent spelling of chaoticity/chaotisity in the introduction should be addressed.

The grammar is corrected.

4) Is it possible to calculate numerical results for s>64 scatterers? Why were results for s=64 scatterers shown only in Figs. 3-4, but not in Figs. 1-2?

Calculations for s>64 are possible but should take a rather long time (the problem complexity scales as s squared). However, I don’t see that such calculations are necessary for verification of the present work conclusions. The non-symmetric and symmetric models approach, respectively, the GUE and GOE prediction at s=32, then the calculations for s>32 are unnecessary for these models. The T-invariant PBC model approaches GOE predictions at s=64. For the HWB model, it was not so clear that it approaches GOE predictions at s=32, and results for s=64, which almost coincide with the s=32 ones, eliminate this doubt.

5) Does there exist an analytical prediction for the spectral rigidity function corresponding to Šeba statistics? As expressed at the bottom of page 6 in the case of Poissonian, GOE or GUE statistics?

In my best knowledge, there are no such predictions. Moreover, Fig. 10 (former 9) demonstrates that Seba NND can correspond to different spectral rigidity plots.

6) Is it possible to realise the Hamiltonian (Eq. (2)) under consideration in experiment? How would one tune the number, position and strength of the scatters, or alter the vector potential/boundary conditions in this case, in order to interpolate between the four specific models that are analysed? Would it be possible to probe the fluctuations in the specific observables considered in Sec. 4?

The experimental realization is discussed in the last paragraph of Sec. 2 in the revised manuscript. The observables are related to the radial coordinate or axial momentum distributions and, therefore, can be observed in experiment.

And a couple of more minor points:

7) There are some inconsistencies regarding the way equations are labelled in both the main text and appendices. For example, following Eq. (18) several equations displayed in the main text remain unlabelled, until Eq. (19) appears are the beginning of Sec. 4. To improve readability, I believe it would be best if all equations appearing in the main text/appendix were labelled in sequence, even those that are not directly referenced in the text.

All equations are numbered in the revised version.

8) In Fig. 7(a), multiple results are plotted with thin black lines. This makes it very difficult to distinguish between these data sets. In general, for plots of the level-spacing distributions presented in Sec. 3, it is often quite difficult to distinguish visually between the different legend entries, and then match these to the appropriate histogram. Perhaps employing some alternate line styles etc. could improve the clarity of these figures.

Fig. 7(a) is corrected. The different line styles are used in Fig. 5(a) with wide histogram bins, but in other figures the bin width is about the dash length, and different styles become indistinguishable.

9) In Fig. 8(a), the third legend entry reads “GOI” rather than GOE.

Corrected.

---

## Round 1 · Referee Report · Anonymous (Referee 2) · 2023-8-23

Strengths

  • It deals with an important topic, the crossover from integrability to chaos in a quantum system.
  • The manuscript is well written.

Weaknesses

  • I approach this manuscript as a follow on to Ref. 54, a recently published Physical Review Letter. This however is not made explicit in the introduction.

Report

This manuscript treats an important topic, that of the crossover from integrability to chaos in a quantum system. The author has chosen a tractable system in which to study this crossover: a quantum mechanical system of a harmonic waveguide with a finite number of scatterers. Four systems of scatterers are considered here with various combinations of boundary conditions and symmetries (parity and time inversion). The symmetries of the scatterers determine how quickly the crossover from integrability to chaos occurs as well as the particular set of statistics governing the energy level spacings. The rapidity with which this crossover occurs is governed by the number of scatterers and their scattering strength.

This manuscript serves as a follow on to a recently published Physical Review Letter (Ref. 54 of the manuscript). My current view of it is that it should be considered a companion article to Ref. 54 where analytical and computational details that would not appear in a letter publication can be spelt out.

Requested changes

  1. The relationship to Ref. 54 should be made explicit in the introduction. I do not think it is a problem for acceptance in Scipost at large that the author explicitly tell the reader that this work provides important details related to results first presented in Ref. 54. It is also an opportunity for the author to explain to the reader what new material does appear here. If there is sufficiently new material as compared to Ref. 54, it appropriately could be published in Scipost Physics. If not, Scipost Physics Core is a more appropriate venue.

  2. The crossover from integrability to chaos is controlled by two knobs: the number of scatterers and strength of scattering. Can the author comment on how these two knobs interact? It would be interesting to understand, for example, how the strength of scattering needed to induce complete chaoticity (as measured by the IPR or closeness of the energy level distribution to GUE or GOE) depends on the number of scatterers in the system.

  • validity: high
  • significance: high
  • originality: high
  • clarity: high
  • formatting: excellent
  • grammar: excellent

Author:  Vladimir Yurovsky  on 2023-09-08  [id 3962]

(in reply to Report 2 on 2023-08-23)

I am indeed grateful to the Referee for careful reading of the manuscript and pointing out unclear statements. All Referee’s remarks are addressed. The revised manuscript contains necessary clarifications, as well as other corrections in response to the Referee's remarks. Following below are replies to all remarks of the Referee.

Weaknesses

I approach this manuscript as a follow on to Ref. 54, a recently published Physical Review Letter. This however is not made explicit in the introduction.

This issue is clarified (see the response to the requested change 1 below).

Requested changes

The relationship to Ref. 54 should be made explicit in the introduction. I do not think it is a problem for acceptance in Scipost at large that the author explicitly tell the reader that this work provides important details related to results first presented in Ref. 54. It is also an opportunity for the author to explain to the reader what new material does appear here. If there is sufficiently new material as compared to Ref. 54, it appropriately could be published in Scipost Physics. If not, Scipost Physics Core is a more appropriate venue.

I am indeed grateful to the referee for pointing out my omission. The relationship to [54] (now [55]) is made explicit in the 4th paragraph of the introduction. Only Fig. 12(a) and several plots in Figs. 12(b) and 13(a) use the same data as figures in [54] (now [55]). Captions to figures 12 and 13 in the revised manuscript specify which data are shared.

The crossover from integrability to chaos is controlled by two knobs: the number of scatterers and strength of scattering. Can the author comment on how these two knobs interact? It would be interesting to understand, for example, how the strength of scattering needed to induce complete chaoticity (as measured by the IPR or closeness of the energy level distribution to GUE or GOE) depends on the number of scatterers in the system.

The interaction between the controlling parameters is illustrated by the new Fig. 8 and by Fig. 12(b) (former Fig. 11), which now includes additional plots for 8 scatterers. These figures demonstrate how the scatterer strength needed to induce the given chaoticity depends on the number of scatterers in the system. Consideration of the case of complete chaos, as proposed by the Referee, requires extensive calculations for s>32 scatterers and various interaction strengths. Such calculations become rather long (the problem complexity increases as the number of scatterers squared) and would be a subject of the following work.

---

## Round 2 · Author Response

I am indeed grateful to the Referees for careful reading of the manuscript and pointing out unclear statements. All their remarks are addressed. The revised manuscript contains necessary clarifications, as well as other corrections in response to the Referee's remarks. Point-by-point response to the remarks is provided in the replies to the referee reports.

---

## Round 2 · List of Changes

Discussion of the relationship to [55] (former [54]) and of the new material appeared in the present work is included in the 4th paragraph of the introduction.
The motivation for investigation of the four particular models is discussed below the paragraph containing Eq. (9).
The complexity of the present model and the possible experimental realization is discussed in the two last paragraphs of Sec. 2.
Figs. 7(a) and 9(a) (former 8(a)) are corrected.
New Fig. 8 is included and discussed.
The level spacing ratio is discussed in the last paragraph of Sec. 3 and in the new appendix C.
Figures 11 and 12 (former 10 and 11) are rearranged and new subfigure 11(b) and additional plots in Fig. 12(b) are included.
Discussion of Figs. 11 and 12 is changed and extended.
Captions to figures 12 and 13 specify which data are shared with [55] (former [54]).
All equations are numbered.
The grammar is corrected.
The motivation for investigation of the four particular models is discussed below the paragraph containing Eq. (9).
The complexity of the present model and the possible experimental realization is discussed in the two last paragraphs of Sec. 2.
Figs. 7(a) and 9(a) (former 8(a)) are corrected.
New Fig. 8 is included and discussed.
The level spacing ratio is discussed in the last paragraph of Sec. 3 and in the new appendix C.
Figures 11 and 12 (former 10 and 11) are rearranged and new subfigure 11(b) and additional plots in Fig. 12(b) are included.
Discussion of Figs. 11 and 12 is changed and extended.
Captions to figures 12 and 13 specify which data are shared with [55] (former [54]).
All equations are numbered.
The grammar is corrected.

---

## Editorial Decision

published